# Group Analysis of the Plane Steady Vortex Submodel of Ideal Gas with Varying Entropy

**Salavat Khabirov** 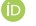

Mavlyutov Institute of Mechanics UFRC RAS, 71 Pr. Oktyabrya, 450054 Ufa, Russia; habirov@anrb.ru

**Abstract:** The submodel of ideal gas motion being invariant with respect to the time translation and the space translation by one direct has 4 integrals in the case of vortex flows with the varying entropy. The system of nonlinear differential equations of the third order with one arbitrary element was obtained for a stream function and a specific volume. This element contains from the state equation and arbitrary functions of the integrals. The equivalent transformations were found for arbitrary element. The problem of the group classification was solved when admitted algebra was expanded for 8 cases of arbitrary element. The optimal systems of dissimilar subalgebras were obtained for the Lie algebras from the group classification. The example of the invariant vortex motion from the point source or sink was done. The regular partial invariant submodel was considered for the 2-dimensional subalgebra. It describes the turn of a vortex flow in the strip and on the plane with asymptotes for the stream line.

**Keywords:** vortex gas flow; varying entropy; group analysis; optimal system of subalgebras; invariant solution; regular partial invariant solution

## 1. Introduction

The model of ideal gas dynamics is studed very good [1–3]. The numerical and analytical methods for solving of the boundary value problems were developed [4,5]. The methods of symmetry (group) analysis were developed for the testing of calculations and detecting new singularities of gas motions [6,7]. The classical results for the plane steady potential flows [2,3,8] were generalized on the vortex isentropic motions [9,10].

As a rule it is not proved the existence and uniqueness of the classical smooth solution as the whole for nonlinear space boundary value problems of the mechanic medium. For the numerical and asymptotic solutions the same it is not proved convergence to the classical solutions of the boundary value problems. Therefore it is value to know possibly more the exact solutions in the enough big domain of space-time continuum. For the classes of exact solutions it is possible more simple submodels. The group analysis makes the classification of these submodels.

In the present paper we consider the mathematical submodel of plane steady vortex flows of the ideal gas with verying entropy for an arbitrary state equation, arbitrary values of the Bernoulli, entropy, vorticity integrals that combined into one arbitrary element. The equivalent transformations of the submodel was obtained by the group analysis methods [1,11–13]. They change only arbitrary element. It was proved the existence of 8 types of the group classification models with differing symmetries. The optimal systems of dissimilar subalgebras admitted by models were constructed. In the fact its give the classification of submodels. The subalgebras produce invariant, partial invariant and differential invariant solutions. The invariant solutions show singularities in the submodel solutions. So it is proved the existence of the plane point source or the sink for vortex entropy steady invariant gas motions in contrast to the plane isentropic potential invariant solution. The example of a regular partial invariant solution was considered on the 2-dimension subalgebra.

## 2. Steady 2-Dimension Submodel and Equivalent Transformations

We consider the gas dynamics equations [8]

$$\vec{u}_t + (\vec{u} \cdot \nabla)\vec{u} + \rho^{-1}\nabla p = 0, \quad \rho_t + (\vec{u} \cdot \nabla)\rho + \rho\nabla \cdot \vec{u} = 0,$$

$$S_t + \vec{u} \cdot \nabla S = 0, \quad p = g(V, S) = -\varepsilon_V(V, S), \quad T = \varepsilon_S, \quad V = \rho^{-1},$$

where $\vec{u}$ is a velocity, $\varepsilon = \varepsilon(V, S)$ is a state equation, $p$ is a pressure, $\rho$ is a density, $\varepsilon$ is a inner energy, $S$ is an entropy, $T$ is a temperature, $V$ is a specific volume are invariant with respect to the translations by time $t$, by space $\vec{x}$, Galilean translations (motion of the origin of coordinates with a constant velocity), the rotations and the proportional dilatation by $t$ and $\vec{x}$. These transformations form 11-parameter group [1]. We consider the invariant motions with respect to the translations by $t$ and $z$ in the Cartesian coordinate system $\vec{x} = (x, y, z), \vec{u} = (u, v, w)$. The invariant steady plane submodel is [2,3,8].

$$Du + Vp_x = 0, \quad Dv + Vp_y = 0, \quad Dw = 0, \quad DS = 0,$$

$$D\rho + \rho(u_x + v_y) = (\rho u)_x + (\rho v)_y = 0. \tag{1}$$

The stream function $\psi(x, y)$ is introduced by the last equation of the system (1)

$$u = V\psi_y, \quad v = -V\psi_x, \quad \partial\psi \equiv 0, \quad D = V\partial, \quad \partial = \psi_y\partial_x - \psi_x\partial_y.$$

With the enthalpy $i = \varepsilon + pV$, $i_V = -V\varepsilon_{VV} = Vg_V$ the system (1) has 3 integrals (Bernoulli, entropy and the third component of the velocity)

$$V^2(\psi_x^2 + \psi_y^2) + 2i = B^2(\psi), \quad S = S(\psi), \quad w = w(\psi). \tag{2}$$

One equation remains from the system (1)

$$(\partial u)_y = (\partial v)_x,$$

which with the help of (2) may be written in the form

$$\partial[-BB'V^{-1} + S'\varepsilon_S V^{-1} + V\triangle\psi + \nabla\psi \cdot \nabla V] = 0.$$

From this it follows the 4th integral which together with the Bernoulli integral form the submodel equations

$$V\triangle\psi + \nabla\psi \cdot \nabla V = \frac{1}{2}K_\psi, \quad \psi_x^2 + \psi_y^2 + K_V = 0, \quad K_V < 0, \tag{3}$$

where $K = B^2(\psi)V^{-1} - 2\varepsilon V^{-1} - 2P_1(\psi)$ is an arbitrary element expressing through the state equation and arbitrary functions of the integrals $B(\psi), S(\psi), P_1(\psi)$.

The velocity curl of the invariant submodel with the help of (2) is equal to

$$\vec{\omega} = \nabla \times \vec{u} = (w_y - v_z, u_z - w_x, v_x - u_y) = (w'\psi_y, -w'\psi_x, \omega), \quad \omega = v_x - u_y.$$

The value $\omega$ by virtue of (3) is equal to

$$\omega = -\frac{1}{2}K_\psi = -P_1' - BB'V^{-1} + S'\varepsilon_S V^{-1},$$

and from (1) and (2) satisfy the equation

$$\partial\omega = -(\omega + S'g_S)V^{-1}\partial V.$$

From here we obtain the vorticity integral

$$\omega V = \Omega'(\psi) + S' \varepsilon_S \quad \Rightarrow \quad K = -2P(\psi) - 2V^{-1}\Omega - 2V^{-1}\varepsilon. \tag{4}$$

Two expressions for $K$ differ on the linear summand by $V^{-1}: P = P_1, 2\Omega = -B^2$.

For isentropic flow the vortex motions was considered in [9,10]. Then the vortex motions with varying entropy will be considered.

For an abitrary element the equations are realized

$$K_x = K_y = 0. \tag{5}$$

The transformations of variables $x, y, V, \psi$ no changing the form of the Equations (3), (5) but changing only the function $K(V, \psi)$ are named the equivalent transformations. These transformations form a group with Lie algebra given by the operators prolonged on the derivatives in Equations (3), (5) [1,12]:

$$Y = \xi^x \partial_x + \xi^y \partial_y + \eta^V \partial_V + \eta^\psi \partial_\psi + \eta^K \partial_K + (\widetilde{D}_x \eta^V - V_x \widetilde{D}_x \xi^x - V_y \widetilde{D}_x \xi^y) \partial_{V_x}$$

$$+ (\widetilde{D}_y \eta^V - V_x \widetilde{D}_y \xi^x - V_y \widetilde{D}_y \xi^y) \partial_{V_y} + \zeta^x \partial_{\psi_x} + \zeta^y \partial_{\psi_y}$$

$$+ (\widetilde{D}_x \zeta^x - \psi_{xx} \widetilde{D}_x \xi^x - \psi_{xy} \widetilde{D}_x \xi^y) \partial_{\psi_{xx}} + (\widetilde{D}_y \zeta^y - \psi_{xy} \widetilde{D}_y \xi^x - \psi_{yy} \widetilde{D}_y \xi^y) \partial_{\psi_{yy}}$$

$$+ (D_j \eta^K - K_x D_j \xi^x - K_y D_j \xi^y - K_V D_j \eta^V - K_\psi D_j \eta^\psi) \partial_{K_j},$$

where $j = x, y, V, \psi$,

$$\zeta^i = \widetilde{D}_i \xi^\psi - \psi_x \widetilde{D}_i \xi^x - \psi_y \widetilde{D}_i \xi^y, \quad i = x, y, \quad D_j = \partial_j + K_j \partial_K,$$

$$\widetilde{D}_i = \partial_i + V_i \partial_V + \psi_i \partial_\psi + (K_i + K_V V_i + K_\psi \psi_i) \partial_K$$

$$+ (K_{ji} + K_{jV} V_i + K_{j\psi} \psi_i) \partial_{K_j} + \psi_{xi} \partial_{\psi_x} + \psi_{yi} \partial_{\psi_y}.$$

The operator coordinates $\xi^i, \eta^l, l = V, \psi, K$ are functions of variables $x, y, V, \psi, K$. The compatibility conditions of the Equations (3), (5) have the form [1]

$$Y((3)) = 0, \quad Y((5)) = 0$$

for the solutions of the Equations (3) and (5). This gives an overdetermined linear system of the homogeneous equations for the coordinates of the operator $Y$.

**Theorem 1.** *The Lie algebra of the equivalent transformations is infinite. The basic operator are*

$$X_1 = \partial_x, \quad X_2 = \partial_y, \quad X_3 = y\partial_x - x\partial_y, \quad X_4 = x\partial_x + y\partial_y + \psi\partial_\psi,$$

$$X_5 = \psi\partial_\psi + 2K\partial_K, \quad <\eta> = -V\eta'(\psi)\partial_V + \eta(\psi)\partial_\psi, \quad <\zeta>_0 = \zeta(\psi)\partial_K,$$

*where $\eta(\psi), \zeta(\psi)$ are arbitrary functions.*

**Proof.** The conditions of invariance for the Equation (5) are

$$0 = YK_x = \eta_x^K - K_V \eta_x^V - K_\psi \eta_x^\psi,$$

$$0 = YK_y = \eta_y^K - K_V \eta_y^V - K_\psi \eta_y^V.$$

We assume that the values $K_V, K_\psi$ are arbitrary. Hence it follows

$$\eta_i^K = \eta_i^V = \eta_i^\psi = 0, \quad i = x, y.$$

The condition of invariance for the first equation of the system (3) may be written in the form

$$
\begin{aligned}
& 2\psi_x[V_x\eta_V^\psi + \psi_x\eta_\psi^\psi - \psi_x(\xi_x^x + V_x\xi_V^x + \psi_x\xi_\psi^x) - \psi_y(\xi_x^y + V_x\xi_V^y + \psi_x\xi_\psi^y) \\
& +(K_V V_x + K_\psi \psi_x)(\eta_K^\psi - \psi_x\xi_K^x - \psi_y\xi_K^y)] \\
& +2\psi_y[V_y\eta_K^\psi + \psi_y\eta_\psi^\psi - \psi_x(\xi_y^x + V_y\xi_V^x + \psi_y\xi_\psi^x) - \psi_y(\xi_y^y + V_y\xi_V^y + \psi_y\xi_\psi^y) \\
& +(K_V V_y + K_\psi \psi_y)(\eta_K^\psi - \psi_x\xi_K^x - \psi_y\xi_K^y)] \\
& +\eta^K - K_V\eta_V^V - K_\psi\eta_V^\psi + K_V(\eta_K^K - K_V\eta_K^V - K_\psi\eta_K^\psi) = 0.
\end{aligned}
\tag{6}
$$

The value $K_\psi$ is proportional a value $\triangle\psi$ which may be arbitrary. The equating to zero of the coefficient under $K_\psi$ in (6) gives

$$
2\psi_x(\psi_x\eta_K^\psi - \psi_x^2\xi_K^x - \psi_x\psi_y\xi_K^y) + 2\psi_y(\psi_y\eta_K^\psi - \psi_x\psi_y\xi_K^x - \psi_y^2\xi_K^y) = \eta_V^\psi - (\psi_x^2 + \psi_y^2)\eta_K^\psi.
$$

The equating to zero of the coefficients under the powers of values $\psi_x, \psi_y$ (the splitting at $\psi_x$ and $\psi_y$) leads to the equations

$$
\xi_K^x = \xi_K^y = \eta_K^\psi = \eta_V^\psi = 0.
$$

The residuals of (6) are the polynomial of 4th power by $\psi_x$ and $\psi_y$. The splitting gives

$$
\begin{aligned}
& \eta_K^V = \eta_V^K = \xi_\psi^x = \xi_\psi^y = \xi_V^x = \xi_V^y = 0, \\
& \xi_x^y + \xi_y^x = 0, \quad \xi_x^x = \xi_y^y = c(x,y), \\
& \eta^K(K,\psi) = \eta_V^V(\psi,V) + 2(\eta_\psi^\psi(\psi) - c) \Rightarrow c = C \text{ is a constant.}
\end{aligned}
$$

From here it follows the presentation for the coordinates of operator $Y$

$$
\begin{aligned}
& \xi^x = Cx + Ey + E_1, \quad \xi^y = Cy - Ex + E_2, \\
& \eta^V = \eta'(\psi)V + \eta_0(\psi), \quad \eta^K = K\zeta_1'(\psi) + \zeta(\psi), \\
& \eta^\psi = \tfrac{1}{2}(\zeta_1 - \eta) + C\psi + D,
\end{aligned}
\tag{7}
$$

where $E, E_1, E_2, D$ are constant.

The condition of invariance for the second equation of the system (3) has the form

$$
\begin{aligned}
& V[\triangle\psi(\eta_\psi^\psi - 2C) + \eta_{\psi\psi}^\psi(\psi_x^2 + \psi_y^2)] + \eta^V\triangle\psi + V_x(\psi_x\eta_\psi^\psi - \psi_x C) \\
& +V_y(\psi_y\eta_\psi^\psi - \psi_y C) + \psi_x(V_x\eta' + \psi_x(\eta''V + \eta_0') - V_x C) \\
& +\psi_y(V_y\eta' + \psi_y(\eta''V + \eta_0') - V_y C) \\
& = \tfrac{1}{2}[K\zeta_1'' + \zeta' + (\psi_x^2 + \psi_y^2)(\eta''V + \eta_0') + (\zeta_1' - \eta_\psi^\psi)(V\triangle\psi + \psi_x V_x + \psi_y V_y)].
\end{aligned}
\tag{8}
$$

The splitting by $\triangle\psi$ and $\psi_x^2 + \psi_y^2$ gives

$$
\eta_0 = 0, \quad \zeta_1'' = 0 \Rightarrow \zeta_1' = M \text{ is a constant}
$$

and (8) is fulfilled identically. The coordinates of operator $Y$ in (7) are corrected

$$
\eta^V = \eta'(\psi)V, \quad \eta^K = MK + \zeta(\psi), \quad \eta^\psi = (C + 2^{-1}M)\psi - 2^{-1}\eta(\psi) + D.
$$

Here and in (7) $\zeta(\psi), \eta(\psi)$ are arbitrary functions, $E, E_1, E_2, C, D, M$ are arbitrary constants. The basis from Theorem 1 is obtained to the equating zero all arbitrary elements except one. $\square$

**Remark 1.** *The transformations no changing the function $K(\psi, V)$ form the kernel of admitting groups $\{X_1, X_2, X_3\}$.*

**Remark 2.** *The transformations changing the function K have the form:*
*(a)* $\widetilde{K} = K + \zeta(\psi)$,
*(b)* $\widetilde{\psi} = b\psi$, $\widetilde{K} = Kb^2$,
*(c)* $\widetilde{x} = cx$, $\widetilde{y} = cy$, $\widetilde{\psi} = c\psi$,
*(d)* $\widetilde{V}\eta(\widetilde{\psi}) = V\eta(\psi)$, $a = \int\limits_{\widetilde{\psi}}^{\psi} \frac{dt}{\eta(t)} \Rightarrow \widetilde{\psi} = \mu(\psi)$, $\mu'\eta(\psi) = \eta(\mu)$, $\widetilde{V} = V(\mu'(\psi))^{-1}$,

*where $\zeta(\psi)$, $\mu(\psi)$ and $\eta(\psi)$ are arbitrary functions; a, b and c are constant group parameters. If $\eta = A$ is a constant then the transformation (d) is the translation by $\psi$*

$$\widetilde{V} = V, \quad \widetilde{\psi} = \psi - aA.$$

*If $\eta(t) = t$ then the transformation (d) is the dilatation*

$$\widetilde{\psi} = \psi e^{-a}, \quad \widetilde{V} = e^a V.$$

**Remark 3.** *The reflection $\psi \to -\psi$ is admitted also.*

### 3. The Group Classification of Submodel

The problem of the group classification consist to find arbitrary elements of the system (3) to within the equivalent transformations for which the admitted group is more than the kernel. The operators of Lie algebra of the point transformations is written in the form prolonged on the derivatives from the Equation (3) [1]

$$\begin{aligned}
X = {}& \xi^x \partial_x + \xi^y \partial_y + \eta^V \partial_V + \eta^\psi \partial_\psi + \zeta^x \partial_{\psi_x} + \zeta^y \partial_{\psi_y} + (D_x \eta^V - V_x D_x \xi^x - V_y D_x \xi^y) \partial_{V_x} \\
& + (D_y \eta^V - V_x D_y \xi^x - V_y D_y \xi^y) \partial_{V_y} (D_x \zeta^x - \psi_{xx} D_x \xi^x - \psi_{xy} D_x \xi^y) \partial_{\psi_{xx}} \\
& + (D_y \zeta^y - \psi_{xy} D_y \xi^x - \psi_{yy} D_y \xi^y) \partial_{\psi_{yy}},
\end{aligned}$$

where $\zeta^i = D_i \eta^\psi - \psi_x D_i \xi^x - \psi_y D_i \xi^y$, $i = x, y$, $D_i = \partial_i + V_i \partial_V + \psi_i \partial_\psi + V_{ij} \partial_{V_j} + \psi_{ij} \partial_{\psi_j}$ are the operator of the full differentiation ($j = x, y$). Here the operator coordinates $\xi^x, \xi^y, \eta^V, \eta^\psi$ are functions of the variables $x, y, V, \psi$. The invariance condition for the first equation of the system (3) has the form

$$\begin{aligned}
& 2\psi_x [\eta_x^\psi + V_x \eta_V^\psi + \psi_x \eta_\psi^\psi - \psi_x (\xi_x^x + \xi_V^x V_x + \xi_\psi^x \psi_x) - \psi_y (\xi_x^y + \xi_V^y V_x + \xi_\psi^y \psi_x)] \\
& + 2\psi_y [\eta_y^\psi + V_y \eta_V^\psi + \psi_y \eta_\psi^\psi - \psi_x (\xi_y^x + \xi_V^x V_y + \xi_\psi^x \psi_y) - \psi_y (\xi_y^y + \xi_V^y V_y + \xi_\psi^y \psi_y)] \\
& + K_{VV} \eta^V + K_{V\psi} \eta^\psi = 0.
\end{aligned}$$

The splitting by the value $V_x \psi_x + V_y \psi_y$ gives

$$\eta_V^\psi = \xi_V^x \psi_x + \xi_V^y \psi_y \quad \Rightarrow \quad \xi_V^x = \xi_V^y = \eta_V^\psi = 0.$$

The change $\psi_y^2 = -\psi_x^2 - K_V$ and the splitting by $\psi_x^2$ and $\psi_x \psi_y$ leads to the equations

$$\xi_x^y + \xi_y^x = 0, \quad \xi_x^x = \xi_y^y = n, \quad n_V = 0.$$

The equating to zero of the coefficients at the linear summands under $\psi_x$ and $\psi_y$ leads to the determining relations

$$\begin{aligned}
& K_V \xi_\psi^x + \eta_x^\psi = 0, \quad K_V \xi_\psi^y + \eta_y^\psi = 0, \\
& 2nK_V + K_{VV} \eta^V + K_{V\psi} \eta^\psi = 0.
\end{aligned}$$

The determining relations are an overdetermined system of equations for an arbitrary element. It was arbitrary if the relations are fulfilled identically for the kernel of the admitted operators. The kernel may be extended for the special functions $K(V, \psi)$. The equivalent transformations may be changed for special classes of arbitrary elements. Here we do not consider of the full classification.

The invariance condition for the second equation of the system (3) with regard for the received relations has the form

$$
\begin{aligned}
\eta^V \triangle \psi + V[(-2n + \eta_\psi^\psi - \underline{\xi_\psi^x \psi_x} - \underline{\xi_\psi^y \psi_y}) \triangle \psi - \xi_\psi^x \psi_{xx} - \xi_\psi^y \psi_{yy} - 2(\xi_\psi^y \psi_x + \xi_\psi^x \psi_y)\psi_{xy} \\
+ \eta_{\psi\psi}^\psi (\psi_x^2 + \psi_y^2)] + (\psi_x V_x + \psi_y V_y)(\eta_V^V + \eta_\psi^\psi - 2n) + \psi_x \eta_x^V + \psi_y \eta_y^V \\
- \psi_x \psi_y (V_x \xi_\psi^y + V_y \xi_\psi^x) + V_x \eta_x^\psi + V_y \eta_y^\psi - \psi_x^2 V_x \xi_\psi^x \\
- \underline{\psi_y^2 V_y \xi_\psi^y} + \overline{(\psi_x^2 + \psi_y^2)(\eta_\psi^V - \underline{V_x \xi_\psi^x} - V_y \xi_\psi^y)} = 2^{-1}(K_{\psi V}\eta^V + K_{\psi\psi}\eta^\psi).
\end{aligned}
$$

Reduction of the underline summands and the equating to zero of the coefficients at $\psi_{xy}, \triangle\psi, \psi_x^2 + \psi_y^2$ gives

$$
\xi_\psi^y = \xi_\psi^x = 0 = \eta_x^\psi = \eta_y^\psi, \quad \eta^V = (2n - \eta_\psi^\psi)V, \quad 0 = \eta_x^V = \eta_y^V, \quad K_{\psi V}\eta^V + K_{\psi\psi}\eta^\psi = 0.
$$

From here it follows $n_x = n_y = 0 = n_\psi \Rightarrow n = N$ is a constant,

$$
\begin{aligned}
\xi^x &= Nx + Ey + E_1, \quad \xi^y = Ny - Ex + E_2, \\
\eta^\psi &= \eta^\psi(\psi), \quad \eta^V = V(2N - \eta_\psi^\psi),
\end{aligned}
$$

where $E, E_1, E_2$ are constants, and 2 determining relations

$$
\begin{aligned}
K_{\psi V}V(2N - \eta_\psi^\psi) + \eta^\psi K_{\psi\psi} &= 0, \\
VK_{VV}(2N - \eta_\psi^\psi) + K_{V\psi}\eta^\psi + 2(N - \eta_\psi^\psi)K_V &= 0.
\end{aligned} \tag{9}
$$

The last equation is integrable by $V$ and the system (9) has the form

$$
\begin{aligned}
V(2N - \eta_\psi^\psi)K_V + \eta^\psi K_\psi - \eta_\psi^\psi K &= \chi(\psi), \\
(VK_V + K)\eta_{\psi\psi}^\psi + \chi'(\psi) &= 0.
\end{aligned} \tag{10}
$$

Here $\chi(\psi)$ is arbitrary function. The determining relations for the function $K(\psi, V)$ give the overdetermined system

$$
\begin{aligned}
V(C - b')K_V + b(\psi)K_\psi - b'K &= \mu(\psi), \\
(VK_V + K)b'' + \mu' &= 0,
\end{aligned} \tag{11}
$$

with some functions $b(\psi), \mu(\psi)$ and a constant $C$.

We must find the general solution of the system (11) to within the equivalent transformations for different $b(\psi)$. If $b'' \neq 0$ then from the second equation of (11) follows

$$
K = -\frac{\mu'}{b''} + \frac{\lambda(\psi)}{V} \sim \frac{\lambda(\psi)}{V}, \quad \lambda = 1 \text{ or } \psi.
$$

Here the equivalent transformations (a) and (d) from subsection 1 act. From the system (11) it is follow to within the equivalent transformation

$$
\mu = 0, \quad C\lambda = b\lambda'.
$$

Substitution $K$ into (10) determines functions $\chi(\psi), \eta^\psi(\psi)$ and $\eta^V$:

$$
\begin{aligned}
\chi = 0, \quad \eta^\psi = \eta(\psi), \quad \eta^V = 2NV\lambda\lambda''(\lambda')^{-2} \quad (C \neq 0); \\
\chi = 0, \quad \eta^\psi = \eta(\psi), \quad \eta^V = -V\eta'(\psi) \quad (C = 0),
\end{aligned}
$$

where $N$ is an arbitrary constant, $\eta(\psi)$ is an arbitrary function.

Next we consider the case $b = B\psi + B_0$. From (11) it is follow $\mu = M$ is a constant,

$$
(C - B)VK_V + (B\psi + B_0)K_\psi = BK + M. \tag{12}
$$

If $B \neq 0$ then the equivalent transformations make $B_0 = M = 0$. General solution of the Equation (12) with the notation $CB^{-1} = m$ is

$$K = \psi k(I), \quad I = V^{-1}\psi^{m-1}$$

for any $m$ and $k' \neq 0$. From corrected Equation (10)

$$\chi' = \psi \eta^{\psi}_{\psi\psi}(Ik' - k),$$
$$(\eta^{\psi}_{\psi} - 2N)\psi Ik' + \eta^{\psi}((m-1)Ik' + k) - \psi \eta^{\psi}_{\psi}k = \chi.$$

it follows

$$m\eta^{\psi}_{\psi} = 2N.$$

If $m = 0$ then $N = 0$ and

$$(Ik' - k)(\psi \eta^{\psi}_{\psi} - \eta^{\psi}) = \chi.$$

At $\psi \eta^{\psi}_{\psi} = \eta^{\psi} \Rightarrow \chi = 0$, $k(I)$ is an arbitrary function,

$$\eta^{\psi} = B\psi, \quad \eta^{V} = -BV.$$

At $\psi \eta^{\psi}_{\psi} \neq \eta^{\psi} \Rightarrow k = -n + K_0 I$ and $K \sim \psi I$, $\eta^{\psi} = \eta(\psi)$ is an arbitrary function, $\eta^{V} = -V\eta'$.

Let $m \neq 0$ then $\eta^{\psi} = 2Nm^{-1}\psi + B_0$, $\chi = M$, $B_0((m-1)Ik' + k) = M$.

If $B_0 = M = 0$ then $k(I)$ is an arbitrary function,

$$\eta^{\psi} = 2Nm^{-1}\psi, \quad \eta^{V} = 2N(1 - m^{-1})V, \quad K = \psi k(I).$$

At $B_0 \neq 0$ the equivalent transformations make

$$K = V^{1/(m-1)}, \quad \eta^{\psi} = 2Nm^{-1}\psi + B_0, \quad \eta^{V} = 2N(1 - m^{-1})V.$$

Case $B = 0$. The Equation (12) has the form

$$CVK_V + B_0 K_{\psi} = M.$$

To within the equivalent transformations we may consider $K = k(I)$, $I = Ve^{m\psi}$, $k' < 0$ at $B_0 \neq 0$ and $K = -\ln V$ at $B_0 = 0$.

Substitution into (10) gives

$$(k + Ik')\eta^{\psi}_{\psi\psi} + \chi' = 0,$$
$$(2N - \eta^{\psi}_{\psi})Ik' + m\eta^{\psi}Ik' - \eta^{\psi}_{\psi}k = \chi \Rightarrow m\eta^{\psi}_{\psi} = 0.$$

Here we may consider that the variables $I, \psi$ are independent. At $m \neq 0$ it follows $\eta^{\psi} = -2N$, $\chi = 0$, $\eta^{V} = 2NV$. At $m = 0$, $k(V)$ is an arbitrary function, $\eta^{\psi} = B$, $N = 0$, $\chi = 0$, $\eta^{V} = 0$. In the case $K = -\ln V$ from (10) it follows $\eta^{\psi} = B$, $\eta^{V} = 2NV$.

Hence it was possible to formulate the following statement.

**Theorem 2.** *The system* (3) *with arbitrary function* $K(\psi, V)$ *admits the kernel* $\{X_1, X_2, X_3\}$ *from the Theorem* 1. *For the special functions there are the following extensions*

1. $K = V^{-1}\lambda(\psi)$, $\lambda' \neq 04$, $X_4 = x\partial_x + y\partial_y + 2\lambda(\lambda')^{-1}\partial_{\psi} + 2\lambda\lambda''(\lambda')^{-2}V\partial_V$;
2. $K = V^{-1}$, $<\eta> = \eta(\psi)\partial_{\psi} - \eta'(\psi)V\partial_V$;
3. $K = \psi k(V\psi)$, $X_4 = \psi\partial_{\psi} - V\partial_V$;
4. $K = \psi k(I)$, $I = V^{-1}\psi^{m-1}$, $X_4 = x\partial_x + y\partial_y + 2m^{-1}\psi\partial_{\psi} + 2(1 - m^{-1})V\partial_V$;
5. $K = k(I)$, $I = Ve^{\psi}$, $k' < 0$, $X_4 = x\partial_x + y\partial_y - 2\partial_{\psi} + 2V\partial_V$;

6.  $K = V^{1/(m-1)}, m \neq 0, \quad X_4 = x\partial_x + y\partial_y + 2m^{-1}\psi\partial_\psi + 2(1 - m^{-1})V\partial_V, X_5 = \partial_\psi;$
7.  $K = k(V), k' < 0, \quad X_5 = \partial_\psi;$
8.  $K = -\ln V, \quad X_4 = x\partial_x + y\partial_y + 2V\partial_V, \quad X_5 = \partial_\psi.$

## 4. Optimal Systems

The Lie algebras of extensions from the Theorem 2 have different dimensions and structures. For the cases $1°, 5°, 4°$ the algebra decompose into the semi-direct sum of the Abelian subalgebra $\{X_3, X_4\}$ and the Abelian ideal $\{X_1, X_2\}$

$$L_4 = \{X_1, X_2\} \oplus \{X_3, X_4\} \tag{13}$$

according to the commutators of the basic operators

$$[X_1, X_2] = 0, \quad [X_1, X_3] = -X_2, \quad [X_1, X_4] = X_1,$$

$$[X_2, X_3] = X_1, \quad [X_2, X_4] = X_2, \quad [X_3, X_4] = 0.$$

The inner automorphisms in $L_4$ are calculated by the rule: for each basic operator $X_k$ the linear transformation is the solution of the following task

$$X'_{a_k} = [X_k, X'], \quad X' = x'_i X_i|_{a_k=0} = X = x_i X_i.$$

For the operator $X_k$ the automorphism $A_k$ is given by transformation of the operator coordinates (it is not written invariable coordinates)

$$\begin{aligned}
&A_1 : x'_1 = x_4 a_1 + x_1, \quad x'_2 = -x_3 a_1 + x_2; \\
&A_2 : x'_1 = x_3 a_2 + x_1, \quad x'_2 = x_4 a_2 + x_2; \\
&A_3 : x'_1 = x_1 \cos a_3 + x_2 \sin a_3, \quad x'_2 = x_1 \sin a_3 - x_2 \cos a_3; \\
&A_4 : x'_1 = x_1 e^{-a_4}, \quad x'_2 = x_2 e^{-a_4}.
\end{aligned}$$

The Abelian subalgebra of the decomposition (13) has the following subalgebras

$$0, \quad X_3 + \alpha X_4, \quad X_4, \quad \{X_3, X_4\}.$$

For each of these subalgebras we add the linear combination from the elements of the Abelian ideal. Some arbitrary coefficients we equate to zero by automorphisms and verify the condition of subalgebra.

We list one-dimension subalgebras to within the automorphisms. To trivial subalgebra we add the linear combination $x_1 X_1 + x_2 X_2$, the automorphism $A_3$ leads to the similar subalgebra $X_1$. Arbitrary subalgebra with the projection $X_3 + \alpha X_4$ is reduced to the projection by the superposition $A_1 A_2$. Similarly the subalgebra $X_4 + x_1 X_1 + x_2 X_2$ is reduced to $X_4$ by $A_1$ and $A_2$. For 2-dimensional subalgebra one from the basic operators may be reduced to one of the listed 1-dimensional subalgebras. For a different basic operator must be realized the condition of the subalgebra: the commutator of them is the linear combination of the basic operators. For example, $[X_3, X_4 + x_1 X_1 + x_2 X_2] = -x_1 X_2 - x_2 X_1 = 0$. From here it follows $x_1 = x_2 = 0$ and we obtain the Abelian subalgebra $\{X_3, X_4\}$. The subalgebra $\{X_4, x_1 X_1 + x_2 X_2\}$ is reduced to $\{X_4, X_1\}$ by the automorphism $A_3$. The condition of the subalgebra for operators $X_3 + \alpha X_4, x_1 X_1 + x_2 X_2$ has the form

$$x_1 X_2 - x_2 X_1 - \alpha(x_1 X_1 + x_2 X_2) = \lambda(x_1 X_1 + x_2 X_2) \Rightarrow x_1 = x_2 = 0.$$

There is no such 2-dimensional subalgebras. There is subalgebra $\{X_1, X_2\}$ with null projection into subspace $\{X_3, X_4\}$. There are no 3-dimensional subalgebras of the type $\{X_3, X_4, x_1 X_1 + x_2 X_2\}$ as the condition of the subalgebra is not realized. There are subalgebras $\{X_4, X_1, X_2\}, \{X_3 + \alpha X_4, X_1, X_2\}$. Hence the optimal system consists of the following

dissimilar subalgebras (*k.i* is number of subalgebra, *k* is subalgebra dimension, *i* is the ordinal number in given dimension)

$$1.1\ X_1, \quad 1.2\ X_3 + \alpha X_4, \quad 1.3\ X_4;$$
$$2.1\ \{X_1, X_2\}, \quad 2.2\ \{X_1, X_4\}, \quad 2.3\ \{X_3, X_4\};$$
$$3.1\ \{X_1, X_2, X_3 + \alpha X_4\}, \quad 3.2\ \{X_1, X_2, X_4\}.$$

For the case $2°$ of the Theorem 2 admitted algebra is infinite. There are the inner automorphisms $A_1, A_2, A_3$. The algebra decompose into the direct sum of 2 ideals

$$\{X_1, X_2, X_3\} \oplus < \eta(\psi) > .$$

The inner automorphisms of 3-dimensional ideal $A_1, A_2, A_3$ calculate subalgebras

$$0, \quad X_3, \quad X_1, \quad \{X_1, X_2, X_3\}.$$

The commutator of operators from infinite ideal is equal to

$$[< \zeta(\psi) >, < \eta(\psi) >] = < \zeta\eta' - \eta\zeta' >$$

The inner automorphism for the operator $< \zeta(\psi) >$ satisfies the problem

$$\bar{\eta}_a = \zeta\bar{\eta}_\psi - \zeta'\bar{\eta}, \quad \bar{\eta}|_{a=0} = \eta(\psi).$$

The solution of this problem has the form

$$\bar{\eta} = \zeta(\psi)G\left(a + \int \frac{d\psi}{\zeta(\psi)}\right), \quad \eta(\psi) = \zeta(\psi)G\left(\int \frac{d\psi}{\zeta(\psi)}\right)$$

The automorphism is given by formula

$$\bar{\eta} = \lambda'(\mu(\psi))\frac{\eta(\lambda(a + \mu(\psi)))}{\lambda'(a + \mu(\psi))},$$

where $\mu(\psi) = \int (\zeta(\psi))^{-1}d\psi$, $\lambda(\mu)$ is inverse function to $\mu(\psi)$. Within this transformation we calculate finite subalgebras in the infinite ideal. The condition for 2-dimensional subalgebras is

$$[< \eta(\psi) >, < \eta_1(\psi) >] = \alpha < \eta(\psi) > + \beta < \eta_1(\psi) > .$$

From this it is follow the equation

$$\eta\eta_1' = \alpha\eta + (\beta + \eta')\eta_1.$$

If $\beta \neq 0$ then $\eta_1 = -\alpha\beta^{-1}\eta + C_0\eta \exp(\beta \int \eta^{-1}d\psi)$ and change of the basis leads to the subalgebra

$$\{< \eta >, < \eta e^{\int \eta^{-1}d\psi} >\}. \tag{14}$$

If $\beta = 0$ then $\eta_1 = C_0\eta + \alpha\eta \int \eta^{-1}d\psi$ and change of the basis leads to the subalgebra

$$\{< \eta >, < \eta \int \eta^{-1}d\psi >\}. \tag{15}$$

We will obtain the 3-dimensional subalgebras using Bianchi classification of the structure over the real field [14]. The structures must not have null commutator. From 2 unsolvable subalgebras is suitable only one with the commutator table of basic elements

$$[X_1, X_2] = X_1, \quad [X_2, X_3] = X_3, \quad [X_1, X_3] = 2X_2.$$

If $X_i = <\eta_i(\psi)>$ then this structure gives the equation system

$$\eta_1\eta_2' - \eta_2\eta_1' = \eta_1, \quad \eta_2\eta_3' - \eta_3\eta_2' = \eta_3, \quad \eta_1\eta_3' - \eta_3\eta_1' = 2\eta_2.$$

The general solution of 2 equations have the form

$$\eta_1 = C\eta_2 e^{-\int \eta_2^{-1}d\psi}, \quad \eta_3 = D\eta_2 e^{\int \eta_2^{-1}d\psi}.$$

The substitution in third equation leads to the relation $CD = 1$.
Thus we obtain the 3-dimensional subalgebra

$$\{<\eta e^{-\int \eta^{-1}d\psi}>, <\eta>, <\eta e^{\int \eta^{-1}d\psi}>\}. \tag{16}$$

The sum of the projections on the ideals gives the subalgebras

$$<\eta>, \ X_1 + <\eta>, \ X_3 + <\eta>, \ \{X_1 + <\eta>, <\eta e^{\int \eta^{-1}d\psi}>\},$$

$$\{X_3 + <\eta>, <\eta e^{\int \eta^{-1}d\psi}>\}, \{X_1, X_2, X_3 + <\eta>\}, (14), (15), (16).$$

For the case $6°$ of the Theorem 2 admitted subalgebra decompose into semi-direct sum of ideal and subalgebra

$$\{X_1, X_2, X_3\} \oplus \{X_4, X_5\}.$$

The automorphisms $A_1, A_2, A_3$ are the same as before, the automorphism $A_4$ has complement $\bar{x}_5' = x_5 \exp(-2a_5)$. There is the new automorphism $A_5 : x_5' = 2x_4a_5 + x_5$. The projections on 2-dimensional subalgebra contain the subalgebras to within the inner automorphisms

$$0, \quad X_4, \quad X_5, \quad \{X_4, X_5\}.$$

Adding projections from the ideal we obtain the optimal system

$$X_1, \quad X_3 + \alpha X_4, \quad X_3 + \alpha X_5, \quad X_4, \quad X_5 + \beta X_1;$$

$$\{X_3, X_4\}, \quad \{X_2, X_4\}, \quad \{X_3, X_5\}, \quad \{X_1, X_5\},$$

$$\{X_4, X_5 + \beta X_1, \beta(m-2) = 0\}, \quad \{X_3 + \alpha X_4, X_5\};$$

$$\{X_4, X_1, X_2\}, \quad \{X_1, X_2, X_3 + \alpha X_4\}, \quad \{X_1, X_2, X_3 + \alpha X_5\}, \quad \{X_1, X_2, X_5\},$$

$$\{X_3, X_4, X_5\}, \quad \{X_4, X_5 + \alpha X_2, X_1, \beta(m-2) = 0\};$$

$$\{X_1, X_2, X_3, X_4\}, \quad \{X_1, X_2, X_3, X_5\}, \quad \{X_4 + \alpha X_3, X_5, X_1, X_2\}.$$

For the case $7°$ of the Theorem 2 the 4-dimensional subalgebra has the center $X_5$. The automorphisms $A_1, A_2, A_3$ produce the optimal system

$$X_1 + X_5, X_3 + \alpha X_5, \alpha = 0 \text{ or } 1; \{X_1, X_2 + \alpha X_5\}, \{X_1, X_5\}, \{X_3, X_5\}; \{X_3 + \alpha X_5, X_1, X_2\}.$$

For the case $8°$ of the Theorem 2 the 5-dimensional subalgebra has the center $X_5$ and the automorphisms $A_1, A_2, A_3, A_4$. The optimal system is similar to the case $4°$ with adding center

$$X_1 + \alpha X_5, X_3 + \beta X_4 + \alpha X_5, X_4 + \alpha X_5; \{X_1 + \alpha X_5, X_2\}, \{X_1 + \alpha X_5, X_4\},$$

$$\{X_3 + \alpha X_5, X_4 + \beta X_5\}, \quad \{X_1, X_5\}, \quad \{X_3, X_5\}, \quad \{X_4, X_5\};$$

$$\{X_1, X_2, X_3 + \beta X_4 + \alpha X_5\}, \{X_1, X_2, X_4 + \alpha X_5\}, \{X_1, X_2, X_5\}, \{X_1, X_4, X_5\}, \{X_3, X_4, X_5\};$$

$$\{X_1, X_2, X_3 + \alpha X_5, X_4 + \beta X_5\}, \{X_1, X_2, X_3 + \alpha X_4, X_5\}, \{X_1, X_2, X_4, X_5\}.$$

The center $X_4$ is added to the kernel for the case $3°$ of the Theorem 2. The optimal system is obtained from the optimal system of the kernel

$$X_4, \ X_1 + \alpha X_4, \ X_3 + \alpha X_4, \ \{X_1, X_2 + \alpha X_4\}, \ \{X_1, X_4\}, \ \{X_3, X_4\}, \ \{X_1, X_2, X_3 + \alpha X_4\}.$$

The optimal system may be presented as the graph of the embedded subalgebras, for example, for the algebra $L_4$ of the case $1°, 4°, 5°$ (Figure 1). The system of embedded subalgebras may be constructed with the help of the graph [15].

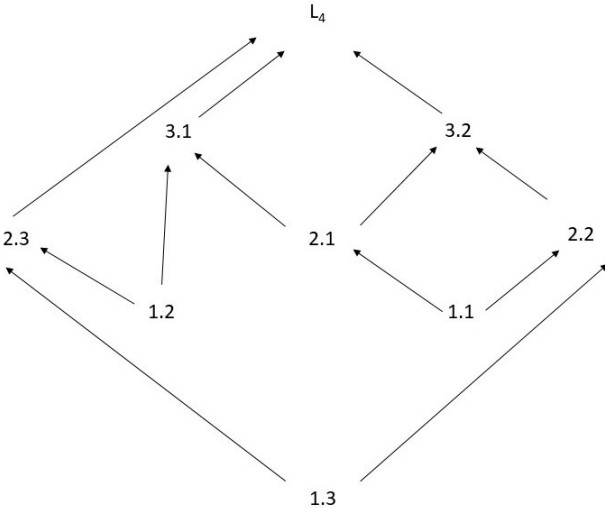

**Figure 1.** The graph of embedded subalgebras.

The constructed optimal systems classify the group submodels of the system (3) in fact. The 1-dimensional subalgebras give the invariant submodels. The 2-dimensional subalgebras give the partial invariant submodels as the simple waves. The subalgebras of big dimensions give the differential invariant submodels with the invariant differential connections.

## 5. The Examples of the Group Solutions

The subalgebra 1.3 of the case $4°$ of the Theorem 2 ($K = \psi k(I)$, $I = V\psi^{1-m}$) determines the invariant solution. It is convenient to use the polar system of coordinates $x = r\cos\varphi$, $y = r\sin\varphi$. The operator of the subalgebra is

$$X_4 = r\partial_r + 2m^{-1}\big(\psi\partial_\psi + (m-1)V\partial_V\big),$$

the Equation (3) have the form

$$\begin{aligned}
\psi_r^2 + r^{-2}\psi_\varphi^2 &= -K_V = -\psi^{2-m}k'(I), \quad k' < 0, \\
V(\psi_{rr} + r^{-2}\psi_{\varphi\varphi} + r^{-1}\psi_r) + \psi_r V_r + r^{-2}\psi_\varphi V_\varphi &= \tfrac{1}{2}K_\psi = \tfrac{1}{2}(k + (1-m)Ik').
\end{aligned} \tag{17}$$

The invariants of the subalgebra give the solution representation

$$\psi = r^{2/m}\Psi(\varphi), \quad V = \psi^{m-1}I(\varphi).$$

The substitution into (17) give the system of the odinary differential equations

$$\begin{aligned}
\Psi'^2 + 4m^{-2}\Psi^2 + \Psi^{2-m}k' &= 0, \\
\Psi'' + 4m^{-2}\Psi + I^{-1}I'\Psi' &= 2^{-1}\Psi^{1-m}(I^{-1}k + (m-1)k').
\end{aligned} \tag{18}$$

We differentiate the first equation and exclude $\Psi''$

$$\Psi I' \left( 8m^{-2}\Psi^m + (Ik)'' \right) + \Psi'(Ik)' = 0.$$

From here we obtain the integral

$$\Psi^{-m} = k_1^m \left( C + 8m^{-1} \int k_1^{-1-m} dI \right), \quad k_1 = (Ik)'.$$

The submodel (18) is integrated in quadratures. On the simple example we consider a behavior of stream lines. Let $C = 0$, $k = DI^n$, $Dn < 0$. Then $k_1 = D(n+1)I^n$ and the integral has the form

$$\Psi^m = 8^{-1}DI^{n-1}m(n+1)(1 - n - nm).$$

The first Equation (18) is

$$\Psi' = \gamma\Psi, \quad \gamma^2 = \frac{4(2n^2 - 1 - nm)}{m^2(n+1)(1 - n - nm)} > 0$$

where the inequality is reached by the choice of $m$ and $n$. Hence the stream function is determined by the equation

$$\psi = Cr^{2/m}e^{\gamma\varphi},$$

where $C$ is constant and the stream line $\varphi = \varphi_0$ is the logarithmic spiral

$$r = (\psi_0 C^{-1})^{m/2} \exp(-2^{-1}m\gamma\varphi).$$

At $m\gamma > 0$, $\varphi \to \infty \Rightarrow r \to 0$. The solution describe the gas motion from the point source or the point sink.

The subalgebra $\{X_1, X_4\}$ for the case $3°$ of the Theorem 2 $K = \psi k(V\psi)$:

$$X_1 = \partial_x, \quad X_4 = \psi\partial_\psi - V\partial_V.$$

The invariants $y$, $I = V\psi$ determine the representation of the regular partial invariant solution of rank 1 and defect 1:

$$V = \psi^{-1}I(y), \quad \Psi = \ln\psi, \quad \Psi = \psi(x, y).$$

The substitution in (3) gives the overdetermined system

$$\Psi_x^2 + \Psi_y^2 = -k'(I), \quad (I\Psi_x)_x + (I\Psi_y)_y = 2^{-1}(k + Ik') = c'(y), \tag{19}$$

where a function $c(y)$ is determined within a constant summand. The change

$$\Psi_x = I^{-1}\chi_y, \quad \Psi_y = I^{-1}(c - \chi_x) \tag{20}$$

satisfies the second equation of (19). The first equation

$$\chi_y^2 + (c - \chi_x)^2 = -I^2k' = b(y)^2$$

is satisfied by the substitution

$$\chi_y = b\cos\vartheta, \quad \chi_x = c - b\sin\vartheta. \tag{21}$$

The compatibility of Equations (20), (21) gives

$$\vartheta_x \sin\vartheta - \vartheta_y \cos\vartheta = b^{-1}(b'\sin\vartheta - c'),$$

$$\vartheta_x \cos \vartheta + \vartheta_y \sin \vartheta = (b^{-1}b' - I^{-1}I') \cos \vartheta.$$

From here the derivatives are determined

$$\vartheta_x = b^{-1}(b' - c' \sin \vartheta) - I^{-1}I' \cos^2 \vartheta, \quad \vartheta_y = (b^{-1}c' - I^{-1}I' \sin \vartheta) \cos \vartheta.$$

The compatibility leads to the relation

$$\left( \frac{I''}{I} - 2\frac{I'b'}{Ib} \right) \sin^2 \vartheta + \left( -\frac{c''}{b} + 2\frac{c'b'}{b^2} + \frac{I'c'}{Ib} \right) \sin \vartheta + \left( \frac{b'}{b} \right)'$$
$$- \left( \frac{I'}{I} \right)' - \left( \frac{c'}{b} \right)^2 + \frac{I'}{I} \left( \frac{b'}{b} - \frac{I'}{I} \right) = 0.$$

From here it follows: either $\vartheta_x = 0$ or all coefficients at the powers $\sin \vartheta$ are equal to zero. At the last case we have the integrals

$$I' = Cb^2, \quad c' = DIb^2,$$

where $C$, $D$ are constants and the equation

$$\left( \frac{I''}{I} \right)' - \frac{I''}{I} - 2D^2C^{-1}I^2I' = 0,$$

which is integrated with the constants $G_0$ and $F_0$

$$I' = \frac{2D^2}{15C}I^5 + 2^{-1}G_0 I^2 + F_0 = -CI^2 k'.$$

From here we find

$$k = -\frac{D^2}{30C^2}I^4 - \frac{G_0}{2C}I + \frac{F_0}{CI} + K_0.$$

The definition $c(y)$ from (19) gives the compatibility condition

$$Ik'(1 + 2DI^2) + k = 0.$$

The substitution the expression $k$ and equating to zero of the coefficients at the power of $I$ gives $D = G_0 = K_0 = 0$. Consequently

$$k = C^{-1}F_0 I^{-1}, \quad I = F_0 y, \quad c = 0.$$

We obtain the compatible system

$$\vartheta_x = -y^{-1} \cos^2 \vartheta, \quad \vartheta_y = -y^{-1} \sin \vartheta \cos \vartheta \Rightarrow \tan \vartheta = -xy^{-1}.$$

Later we solve the system (20)

$$\Psi = \gamma \ln \left| \frac{x}{y} + \sqrt{1 + \frac{x^2}{y^2}} \right|, \gamma = -\sqrt{\frac{F_0}{C}} \Rightarrow \psi = \left| \frac{x}{y} + \sqrt{1 + \frac{x^2}{y^2}} \right|^\gamma.$$

The stream lines $\psi = \psi_0$ are the rays $x = k_0 y$. Along a stream line the density $\rho = V^{-1} = \psi(F_0 y)^{-1}$ is infinite at the origin and it is vacuum at infinity.

For the different case of alternative $\vartheta = \vartheta(y)$ it follows from (20)

$$\chi = \chi_0(y), \quad c = b \sin \vartheta, \quad \chi_0' = b \cos \vartheta = C_0 I,$$

$$\Psi = C_0 x + \Psi_0(y), \quad c^2 + I^2(C_0^2 + k') = 0, \quad I\Psi_0' = c.$$

From the definition function $c(y)$, (19) it follows

$$dy = -\frac{2C_0^2 + (Ik)''}{(IK)'(-C_0^2 - k')^{1/2}} dI$$

and $\Psi_0(y)$ is determined by the expression

$$(Ik)'\Psi_{0I} + 2C_0^2 + (Ik)'' = 0.$$

Hence the solution is determined by the given function $k(I)$.

**Example 1.** *Let $k = I^{-1}$, $\vartheta(y) \Rightarrow c = 0$, $b = 1$. From (21), (20) it follows $\vartheta = \vartheta_0$, $I = I_0$ are constants,*

$$\Psi = \ln \psi = I_0^{-1}(x \cos \vartheta_0 + y \sin \vartheta_0).$$

*The stream lines $\psi = \psi_0$ are straight lines.*

**Example 2.** *Let $k = -C_0^2 I + I^{-n}$, $n > 0$. Then $c^2 = nI^{1-n}$, $J^2 = I^{-1-n}$*

$$-\frac{n+1}{2\sqrt{n}} dy = \frac{dJ}{J^2 + 2C_0^2(n-1)^{-1}}, \quad \Psi_{0I} = \frac{n(1-n)I^{-n-2}}{-2C_0^2 + (1-n)I^{-n-1}}.$$

*The integrating gives the formulas*

$$\Psi_0 = -\frac{n}{n+1}\ln\left| J^2 + \frac{2C_0^2}{n-1} \right|,$$

$$J = \frac{\sqrt{2}C_0}{\sqrt{n-1}}\tan\left(-\frac{n+1}{\sqrt{2n(n-1)}}C_0 y\right) \quad (n > 1);$$

$$J = \frac{\sqrt{2}C_0}{\sqrt{1-n}}\tanh\left(-\frac{n+1}{\sqrt{2n(1-n)}}C_0 y\right) \quad (n < 1).$$

*The stream function is determined by the equation within the constant summand*

$$\ln \psi = C_0 x + 2n(n+1)^{-1}\ln|\cos y_1|, \quad (n > 1);$$

$$\ln \psi = C_0 x + 2n(n+1)^{-1}\ln|\cosh y_1|, \quad (n < 1), \quad \sqrt{2n|n-1|}y_1 = -(n+1)C_0 y.$$

*The stream line $\psi = \psi_0$ is determined by the equations*

$$\cos y = e^{-x} \ (n > 1), \quad \cosh y = e^{-x} \ (n < 1)$$

*within the translation on $x$ and dilatation on $x$ and on $y$. It is even with respect to $y$ and by translation on $x$ cover the flow domain. At $n > 1$ the stream lines give the turn back of the flow in the strip $|y| < \pi/2$ (Figure 2a). At $n < 1$ we obtain the turn of the flow on the plane with the asymptotes $y = \pm x + \beta$, $\cosh \beta = 5/4$ for the stream lines (Figure 2b).*

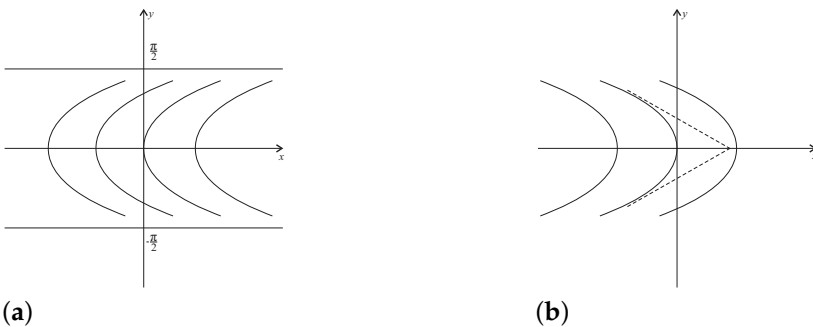

(**a**)                                     (**b**)

**Figure 2.** (**a**) The turn of a flow in the strip. (**b**) The turn of a flow on the plane with asymptotes.

## 6. Conclusions and Discussion

In the present paper we made the symmetry analysis of the steady plane vortex submodel for the ideal gas flow with varying entropy. With the help of 4 integrals the submodel is given by nonlinear system of the third order differential equations for the stream function and the specific volume. In this system there is one arbitrary function on 2 variables which is expressed through the state equation and arbitrary functions of the integrals. We found all equivalent transformations, listed arbitrary elements for which the admitted group is extended. We constructed the optimal systems of subgroups for the each of these extensions. The optimal systems classify group submodels. The examples of the invariant and regular partial invariant solutions were done.

Classification of the group solutions is not completed. There are only several solutions for which the gas particles motion was investigated. The gas motion has its specific for each subalgebra. The determination of these specific characters is not solved problem.

**Funding:** The author was supported by the Russian Foundation for Basic Research (project no. 18-29-10071) and partially from the Federal Budget by the State Target (project no. 0246-2019-0052).

**Conflicts of Interest:** The author declares no conflict of interest.

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
