# Peer review of "Group Analysis of the Plane Steady Vortex Submodel of Ideal Gas with Varying Entropy"

_mathematics, doi:10.3390/math9162006_

Round 1

Reviewer 1 Report

In this manuscript the author introduces a mathematical submodel of plane steady

vortex flow of ideal gas with varying entropy and an arbitrary state function.

After having introduced the stream function the author performs an accurate symmetry classification and gives some examples of group solutions The results are of certain interest and obtained , mathematically speaking , in an elegant way. The paper can be accepted for publications after an accurate revision of the English form and a revision of the section in order to improve the clarity of the description of governing equations.

Author Response

  1. The English language has been improved.
  2. The determining relations are an overdeterming system of equations for an arbitrary element. It was arbitrary if the relations are fulfilled identically for the kernel of the admitted operators. The kernel may be extended for the spacial functins $K(V,\psi)$. 
  3. The equivalent transformations may be changed for spacial classes of arbitrary elements. Here we do not consider of the full classification. 

Reviewer 2 Report

The manuscript contains sufficient work which is interesting. It justifies a new publication. However some points must be cleared by the author before publication.

--- He considers a submodel.  Can the results presented here be obtained from the original model?

--- "Equivalence transformations were found for arbitrary elements" stated by the author. In the equivalence transformations, the arbitrary elements are arbitrary.

---In thm1 needs to be stated the number of the system under study.

--- English text must be significantly improved.

Author Response

  1. The original model is less symmetrical. Therefore, not all results can be obtained using point transformations. Introducing of potential is not a point transformation. 
  2. Equivalence transformations act in the space of arbitrary elements. If arbitrary elements from a certain class of functions, then equivalence transformations choose the simplest representative in this class. 
  3. The number of the system was indicated in Theorem 1. 
  4. The English language was improved.